# HYSTEROSCOPIC MYOMECTOMY

**DOI:** 10.3390/medicina58111627

**Published:** 2022-11-11

**Authors:** Ricardo Bassil Lasmar, Bernardo Portugal Lasmar, Nash S. Moawad

**Affiliations:** 1Department of Surgery and Specialized, Faculty of Medicine, Universidade Federal Fluminense, UFF, Niterói 24020-141, RJ, Brazil; 2University of the Maternal-Infant Department of the Faculty of Medicine, Universidade Federal Fluminense, UFF, Niterói 24020-141, RJ, Brazil; 3Estácio de Sá University, São João de Meriti 25550-100, RJ, Brazil; 4Gynecological Endoscopy, Hospital Central Aristarcho Pessoa HCAP–CBMERJ, Rio de Janeiro 20261-243, RJ, Brazil; 5Division of Minimally Invasive Gynecologic Surgery, Department of Obstetrics & Gynecology, P.O. Box 100294, Gainesville, FL 32610, USA; 6UF Health COEMIG, P.O. Box 100294, Gainesville, FL 32610, USA; 7University of Florida College of Medicine, P.O. Box 100294, Gainesville, FL 32610, USA

**Keywords:** leiomyomas, hysteroscopic myomectomy, submucosal, fibroids

## Abstract

Leiomyomas are the most common pelvic tumors. Submucosal fibroids are a common cause of abnormal bleeding and infertility. Hysteroscopic myomectomy is the definitive management of symptomatic submucosal fibroids, with high efficacy and safety. Several techniques have been introduced over time and will be covered in depth in this manuscript. Advances in optics, fluid management, electrosurgery, smaller diameter scopes, and tissue removal systems, along with improved training have contributed to improving the safety and efficiency of hysteroscopic myomectomy.

## 1. Introduction

Direct visualization of the uterine cavity via hysteroscopy for diagnosis and management is essential for the care of all women with abnormal uterine bleeding, infertility, and suspected intra-uterine pathology. The technologic advances in optics, scope diameters, fluid management, tissue removal systems, and electrosurgery have enabled this minimally invasive approach for the conservative management of intra-uterine pathology and has expanded the ability to perform hysteroscopy in the outpatient setting, making many blind procedures less favorable [1]. This manuscript will focus on advances in hysteroscopic myomectomy for the safe and effective management of submucosal myoma.

### 1.1. Leiomyomas

Uterine leiomyomas arise from the myometrial smooth muscle and the fibroblasts. The immature cells of the myometrium are stimulated by the upregulation of the steroid receptors, leading to the growth of leiomyomas [2,3]. Studies have also shown increased aromatase activity in leiomyoma tissues, leading to increased growth and development [4]. 

The growth or regression rate of myoma size varies significantly. Peddada et al., on a premenopausal women survey, found that the myomas varied widely in their growth rates; they ranged from shrinkage of 89% to growth of 138% per 6 months. The median fibroid growth rate for both black and white women was 9% per 6 months. Solitary myomas appear to grow faster than multiple myomas [5]. 

### 1.2. Epidemiology

Fibroids are the most common tumor diagnosed in the reproductive organs. However, prevalence cannot be accurately assessed due to the asymptomatic and underdiagnosed nature in many patients. Fibroid tumors were found in 77 of 100 uteri after hysterectomy and 84% of the specimens contained multiple fibroids, supporting the extremely high prevalence in most women [6].

African American women have been shown to have fibroid tumors more frequently than Caucasian women. Studies have found increased levels of aromatase mRNA in the leiomyoma tissue of African American females [7]. The rate of growth and the likelihood of rapid expansion of a fibroid decrease with age in Caucasian women but not in African American women [5]. 

Family history, as well as obesity, alcohol intake, soybean product consumption, red meat consumption, hypertension, and vitamin D deficiency have been associated with increased prevalence of uterine fibroids [8,9,10,11]. 

### 1.3. Symptomatology

Many patients with fibroid tumors can be asymptomatic and incidentally diagnosed, and they should be reassured about the benign nature of fibroids in most cases and educated about their trajectory, possible symptoms, treatment options, and outcomes, along with red flags that should prompt additional care. 

Approximately 70% of uterine fibroids lead to abnormal uterine bleeding, which is the most common indication for hysteroscopic myomectomy. Submucosal fibroids are the most implicated in abnormal uterine bleeding [12]. Plausible explanations include distortion of the uterine cavity and increase in the endometrial surface area. In addition, contractility of the myometrium can be impaired by the intervening fibroids [13]. Hysteroscopic resection of leiomyomas provides symptomatic relief in 70–99% of cases [14]. 

Other indications for hysteroscopic myomectomy include subfertility, dysmenorrhea, and pelvic pain [15]. The American Society for Reproductive Medicine currently states that resection should take place for cavity-distorting myomas to improve pregnancy rates and decrease risk of early pregnancy loss [16].

## 2. Preoperative Assessment

Patients with uterine fibroids may complain of abnormal uterine bleeding, infertility, or discomfort from compression of other organs or increased abdominal girth.

Submucosal fibroids are more related to abnormal uterine bleeding and infertility, as they are usually symptomatic even before reaching large volumes [17]. In the presence of bleeding, other causes of it should be investigated, from hematological, functional, or neoplastic causes.

The surgical procedure can be conservative or not of the uterine matrix; for this, it will be important to counsel the patient and evaluate the cases well so that the complexity of the conservative surgery, the myomectomy, can be evaluated.

During initial assessment, the history will bring up important information about the issues mentioned above and the desire for future pregnancy, which would lead to consideration of conservative surgery.

The physical examination, especially the bimanual exam, will provide information about the dimensions and presence of other uterine fibroids. Upon exam, the perception of intramural or subserous myoma, as well as submucous myomas, should prompt transvaginal ultrasound, as the classification (LASMAR) is based on the total volume of the nodules [18].

A proper physical examination is essential to rule out other causes for AUB. Vaginal atrophy should be addressed and gross lesions of the cervix and the vagina, such as polyps or a prolapsed myoma, should be evaluated [11]. During the physical examination, patients’ tolerance to exam is assessed to aid in decision making of whether they can be candidates for office hysteroscopy. 

Patients with AUB should undergo a complete blood count [19]. It is appropriate to evaluate kidney function and consider imaging the kidneys and ureters if fibroids are felt to impact the urinary tract. A pregnancy test should be performed for all reproductive-age women presenting with abnormal bleeding and before any intra-uterine procedures. Infertility specialists should be part of the evaluation for patients with fibroids who have been trying to conceive.

Methods that investigate the submucous myoma in the uterine cavity are more accurate in relation to the myoma, confirming its presence, number, location, and correlation with the myometrium. These are hysterosalpingography and hysteroscopy.

The methods that allow visualization of the uterine cavity and the entire uterine wall are transvaginal and pelvic ultrasound, hysterosonography, and MRI of the pelvis.

Hysterosalpingography has the advantage, in patients with infertility, of concomitant evaluation of tubal patency and configuration. It signals the presence of myoma, but its location in relation to the myometrium is not efficient.

Hysteroscopy, as a method of direct visualization of the uterine cavity, offers all possible information about the intracavitary portion of the submucous myoma and a good assessment of the portion of the myoma, which is found in the myometrium, intramural portion. Thus, with hysteroscopy, it is possible to classify the submucous myoma and assess the need for other imaging methods. Another important function of hysteroscopy is to rule out other intrauterine causes of bleeding and to carry out an anatomopathological study of the endometrium or of the identified lesions, so it should, whenever possible, be indicated in the investigation (Figure 1).

Ultrasonography (USG), especially transvaginal ultrasound (TVUS), is the routine exam and is usually the first one performed; it has good accuracy, easy access, and low cost, but it has a limited role in the presence of a large uterus or multiple nodules, as posterior acoustic shadowing makes it difficult to evaluate and count them. It is important in the evaluation of the intramural component of the myoma and the free myometrial mantle up to the serosa, but it is operator-dependent (Figure 2).

Hysterosonography, an ultrasound procedure performed with the uterus distended with saline solution for greater contrast and detailing of the uterine cavity, is more accurate than TVUS in identifying the uterine cavity and myometrial mantle (Figure 3).

Magnetic resonance imaging of the pelvis (MRI) is indicated in uteri with a volume greater than 375 cm^3^ or with more than four fibroids [20]. With excellent definition regarding the number, location, size of nodules, and proximity to other myomas, it is used to diagnose adenomyosis and adenomyoma, rule out non-fibroids and sarcomas, and to measure the myometrial mantle. The myometrial mantle refers to the distance between the deepest portion of the myoma in the myometrium and the serosa, being of unique importance in hysteroscopic myomectomy, since confirmation of transmural myoma (the one that reaches the serosa) contraindicates the hysteroscopic approach due to the high probability of uterine perforation during the procedure (Figure 4).

OFFICE HYSTEROSCOPY—Significant advances have been introduced to facilitate office hysteroscopy for diagnostic and therapeutic purposes, such as smaller diameter scopes, flexible hysteroscopes, the miniresectoscope, and the tissue removal systems. This can be very valuable for surgical planning and patient education. Small, 1 to 2 cm type 0 submucosal myomas can potentially be removed in the office setting using hysteroscopic scissors or tissue removal systems. A prospective study of patient outcomes after hysteroscopic myomectomy found higher successful completion rates when the fibroids were up to 3 cm in size [21]. However, this size may be difficult for patients undergoing an office procedure. Incision of the pseudocapsule during office hysteroscopy may allow the protrusion of the fibroid into the uterine cavity, improving the likelihood of complete resection during subsequent hysteroscopic myomectomy [21]. 

## 3. Preoperative Classification

As hysteroscopic myomectomy is performed within the uterine cavity (limits of movement and approach), it needs a liquid medium to distend it (risk of intravasation) and, as it often advances into the myometrium (risk of bleeding and intravasation), prior assessment of the difficulty and possibility of hysteroscopic myomectomy is crucial. In addition to the surgeon’s experience and the necessary instruments, and the patient’s clinical conditions, fibroid classification is essential to minimize risks.

The classification of submucous myoma, standardizing it in levels, allows us to indicate the degree of difficulty and complexity of hysteroscopic myomectomy and the comparison of results. There are currently two main classifications: the ESGE, described by Wansteker et al. in 1993 [22], and the Lasmar—STEP-W, published in 2005 [18] (Table 1 and Table 2).

The ESGE classification describes submucosal fibroids in three levels: level 0 = completely in the uterine cavity; level 1 = with its largest portion inside the uterine cavity; and level 2 = with its smallest portion in the uterine cavity. The Lasmar classification evaluates five parameters: nodule size, topography, extension of the base in relation to the affected wall, penetration into the myometrium, and affected wall, to signal the possibility, complexity, or impossibility of hysteroscopic surgery.

How to evaluate each parameter of the Lasmar classification:

**Size of the nodule. (SIZE)**—It is the largest diameter of the myoma identified in one of the imaging tests. When the nodule measures up to 2 cm, it receives a score of 0; between 2 and 5 cm receives score 1, and measuring more than 5 cm receives score 2.

**Location—(TOPOGRAPHY)**—It is determined by the third of the uterine cavity where the myoma is located, with a score of 0 when it is located in the lower third, a score of 1 in the middle, and a score of 2 in the upper third.

**Extension of the myoma base** in relation to the affected wall (**EXTENSION**)—When the myoma base affects 1/3 or less of the uterine wall, it receives a score of 0; when the base of the nodule occupies 1/3 to 2/3 of the wall, the score is 1 and, when it affects more than 2/3 of the wall, the score is 2.

**Penetration** into the myometrium (**PENETRATION**) follows the same principle as ESGE in relation to penetration of the myoma into the myometrium: scores 0, 1, and 2.

**Uterine wall** (**WALL**)—Myoma of the anterior and posterior wall receives a score of 0, while the one located on the lateral wall scores 1.

Before the hysteroscopic myomectomy, other evaluations are important for the surgical procedure: the clinical evaluation of the patient, mainly blood count and coagulogram, since most of them have AUB; and the desire for a future pregnancy, due to the possibility of extensive surgeries and, consequently, uterine adhesions.

According to the Consensus Statement from the Global Congress on Hysteroscopy Scientific Committee, transvaginal ultrasound should be the first line to evaluate the number, size, and location of submucous myomas, with subsequent in-office hysteroscopy to allow, when feasible, a see-and-treat approach [23].

Hospital myomectomy will be indicated if outpatient myomectomy was not possible, determined from the patient’s nontolerance to pain, lack of resources and qualification of the hysteroscopy specialist, and, mainly, the classification of the myoma.

The effect of nearby fibroids, represented by a submucous fibroid with another intramural fibroid next to it, is considered. In this case, the Lasmar classification starts to consider the set as a single node, thus changing the final classification. This is due to the complexity, risk, and surgical result, since, when the myomectomy of the submucous myoma is concluded, another intramural fibroid will be found, which will probably be close to the serosa or have a subserosal component (Figure 5).

Thus, with the new classification, it will be possible to modify the surgical approach for myomectomy. In these cases, hysteroscopic myomectomy associated with laparoscopic myomectomy is indicated.

### 3.1. Hysteroscopic Myomectomy Techniques

Myomectomy, whether laparotomic or laparoscopic, is a well-established procedure, widely performed with the goal of uterine preservation. In both approaches, the myomectomy technique is the same: incision of the serosa up to the pseudocapsule, identification of the myoma, traction and movement of the nodule, assistance in dissecting the plane of the pseudocapsule within the myometrium, and enucleation of the myoma from the uterine wall. This fibroid enucleation technique is known and performed by all gynecologists. When the pseudocapsule is reached, the chance of preserving the uterus will be greater, with less bleeding and less myometrial damage, which differs from adenomyosis resection, which does not have a pseudocapsule [21] (Figure 6).

The presentation of the techniques will make the presentation of this text more didactic, as we basically have two techniques, which can be associated or isolated, each one having its own indication of excellence. These are the enucleation technique and the slicing or myolysis technique, all of which can be performed in outpatient and inpatient hysteroscopy. The enucleation technique was described by Mazzon in 1995 [24] and Lasmar in 2001 [25]. Both techniques have the same basis for enucleation of the nodule, but Mazzon fragments the nodule until it reaches its intramural portion and then uses a “cold loop” to mobilize the fibroid, while Lasmar enucleates the entire fibroid and then slices it. The technique is to incise the endometrium around the submucosal myoma to reach the pseudocapsule (Lasmar) or to reach this plane by slicing the myoma close to the myometrium (Mazzon). Arriving at the pseudocapsule, some fibrous beams must be sectioned. The mobilization of the myoma, from the outside to the center, from front to back, progressively frees it from the myometrium, without significant bleeding and without thermal damage, with a lower risk of intravasation, as it does not cut the myometrial vessels. The mobilization of the myoma with its enucleation can be performed by all instruments, without energy, the use of scissors or tweezers being more appropriate in the outpatient clinic and Collins loop or “cold loop” in hospital hysteroscopy. The slicing technique is based on the progressive cutting of the submucosal portion of the myoma, maintaining the fragmentation of the intramural portion, leading, in most cases, to greater removal of the endometrium and myometrium, with greater thermal damage and risk of intravasation [26]. Fragmentation of the myoma can be performed with a semi-circle loop, with mono or bipolar energy, LASER fiber or morcellator. Thus, the technique of excellence in hysteroscopic myomectomy is the enucleation of the intramural portion of the submucosal myoma, mobilizing the nodule and separating it from the wall of the uterus, while fragmentation would deal with the removal of the myoma from the uterine cavity. 

### 3.2. Outpatient Hysteroscopic Myomectomy 

Outpatient hysteroscopic myomectomy is a safe procedure, immediately treating the lesion at the same time as diagnosis, reducing the patient’s concern and anxiety, as well as complaints. It has a lower cost compared to surgery in a hospital environment and, for the hysteroscopic surgeon, the pleasure of performing the best technique and art of hysteroscopy. However, there are limits to be respected. The limits for outpatient hysteroscopic myomectomy depend on some factors, which, when combined, increase the difficulty of performing the procedure. These are related to the patient, the fibroid, the applied technology, and the hysteroscope.

As for the patient, the main limiting factor is her sensitivity to discomfort, which may allow for a diagnostic examination but precludes outpatient surgery.

The size of the fibroid, its location, fundic or cornual, and the greater penetration into the myometrium are determining factors to hinder or prevent an outpatient myomectomy (Lasmar classification). The combination of factors increases the difficulties in performing an outpatient myomectomy [21].

The instruments used and the type of energy can increase the possibility of performing outpatient surgery. Myoma mobilization is more useful in fibroids with greater penetration into the myometrium, while morcellation techniques are favorable in larger fibroids. Fundic and cornual fibroids can be challenging regardless of the technique.

The experience of the hysteroscopist is crucial for performing an outpatient hysteroscopic myomectomy.

For those new to outpatient surgery, it is advisable to start hysteroscopic myomectomy in the smallest fibroids, 1 to 2 cm, entirely in the uterine cavity, not worrying about immediate extraction or late expulsion of the nodule.

In our service, the most performed technique is using the 5 Fr tweezers or scissors. Initially, the endometrium is incised around the nodule until accessing the plane of the pseudocapsule; then, with the forceps or the body of the hysteroscope, entering between the nodule and the myometrium, the release is initially performed, laterally first and then centrally, until its complete release (Figure 7).

At the end, the nodule will be loose in the cavity and can be fragmented or completely removed with grasping forceps. In cases of difficulty in removing the nodule from the cavity, the patient should be instructed to return in 7 to 10 days, during which time either the nodule will be spontaneously expelled by the patient—she should be oriented about this possibility—or it will have drastically decreased in size, allowing its removal.

When using instruments with energy, we can use the bipolar Collins loop of a miniresectosope system [26] (Figure 8) or the LASER fiber to incise the endometrium around the myoma. However, all mobilization is performed mechanically with forceps, a loop, or the resectoscope itself.

The size of the fibroid can make it difficult to approach the base of the nodule. Larger fibroids can be resected more easily using an energized loop, which allows for a reduction in the nodule and greater ease of approaching the base for mobilization.

When performing an outpatient myomectomy, frequently, the nodule is larger than the internal os, making it impossible to remove it from the uterine cavity at the time of procedure. As mentioned before, it is safe to leave the nodule in the cavity. In the availability of a resectoscope, LASER, or morcellator, the slicing of the lesion is performed with its complete removal.

### 3.3. Hospital Hysteroscopic Myomectomy

Hospital myomectomy is a procedure in which the patient is also assisted by the anesthesiologist in a hospital environment. It is indicated when the myoma classification signals a complex hysteroscopic myomectomy, in patients with low tolerance to the outpatient procedure, and when the hysteroscopist does not have instruments or experience in outpatient myomectomy. Compared with outpatient myomectomy, hospital myomectomy generally has a longer operative time, with the possibility of bleeding and intravasation, risks inherent to complex myomectomy with a more difficult approach, and, therefore, should be performed under anesthesia and in a surgical center [26].

The advantages of hospital myomectomy, in addition to the patient not feeling any discomfort or pain, are: safety in patient monitoring, and bleeding and fluid balance control. This control is essential, as these are the cases of greater complexity and risk of complications. Hospital hysteroscopic myomectomy is a highly complex procedure, being associated with the risks of bleeding, uterine perforation, incomplete surgery, pelvic organ injuries, and intravasation [20].

Anesthesia can be sedation in myomectomies with shorter operative time and spinal block in those with a longer time, so that there is greater control of the patient’s level of consciousness and less use of medications. In this way, each surgical team will decide the type of anesthesia according to the technique and technology used, operative time, surgeon’s experience, and complexity of the case. As previously reported, myomectomy can be divided into fibroid enucleation and fragmentation or myolysis of the fibroid, without the use of energy or with different energy modalities.

It is important to emphasize that the technique and systems influence the complete removal or not of the myoma, but two factors are decisive: the surgeon’s experience and the classification of the myoma.

Even in the hospital environment, the use of scissors or tweezers can also be effective, especially in smaller and more intracavitary fibroids. The technique is the same as described for outpatient myomectomy: access to the pseudocapsule and mobilization of the base with the enucleation. This is a simple technique and does not need dilation of the cervix, just the operative canal and good training for ambulatory operative hysteroscopy [26].

For the introduction of the resectoscope, dilation of the cervix is frequently necessary, except for the miniresector, with its 16 Fr diameter, which can be attached to the same hysteroscope for diagnosis.

The technique with the resectoscope, regardless of the type of energy, is the same, with planned loop movements always in the fundus–cervical direction, with the angulation of the resectoscope axis to define the degree of resection depth. These two movements have to be thought out and prepared before activating the energy so that only the myoma is resected, avoiding resection of the myometrium and the risk of perforation, and so that the penetration of the cut is as desired, without risk (Figure 9).

The movement of the resection loop with energy can only be moved in the fundus–cervix direction, but, without energy, it can be driven in any direction, even the cervix–fundus, as it will have mechanical action. This nonenergy loop movement, called a cold loop, is often used to mobilize and enucleate the submucosal fibroid.

### 3.4. Slicing Technique

The principle of the slicing or slicing technique is the partial and progressive removal of the myoma, in fragments, starting at its surface and gradually working towards its base. Slices of myoma are removed with the semicircle loop in the mono or bipolar resectoscope, moving it energized, from the fundus to the cervix. The distension medium is different according to the type of energy; with monopolar energy, non-electrolytic media are used, which are 1.5% glycine, mannitol, and mannitol/sorbitol, while, with bipolar energy, the electrolytic media, physiological solute 0.9%, and ringer lactate are used [27] (Figure 10).

Due to the crowding of myoma fragments in the uterine cavity, it is necessary to interrupt the procedure with emptying of the cavity, so that the vision of the cavity and the myoma is recovered.

It has the advantage of being able to surgically treat larger nodules, removing the myoma fragments from the cavity, performing volumetric reduction, and performing hemostasis at the same time. As a disadvantage, there is greater bleeding in the procedure (the myoma vessels are superficial), greater possibility of intravasation, especially in myomas with a greater intramural component, greater risk of perforation, and frequent interruption of surgery to remove fragments, in addition to greater endometrial and myometrial damage adjacent to the myoma.

There is a possibility of incomplete myomectomy because, when there are signs of massive fluid absorption and risk of intravasation syndrome, or long operative time and risk of perforation, the procedure is interrupted for a new one around three months later.

The regulation of the electrosurgical generator is cut-coagulation and blend determined by the surgeon’s need for each case and according to each generator, varying cut from 60 to 120 W and coagulation from 40 to 60 W.

Remember that the speed of movement of the loop can also determine the action of more cutting or more coagulation; the faster moving loop cuts more and coagulates less, while the slower one coagulates more than it cuts.

### 3.5. Morcellator Technique (Hysteroscopic Mechanical Tissue Removal)

Hysteroscopic Tissue Removal Systems (TRS) perform fragmentation and suction of endometrial pathology, such as polyps and fibroids. There are three main brands currently available on the market (e.g., Myosure, Truclear, and Symphion) and they are mainly used for types 0 and 1 intrauterine leiomyomas. A rapidly rotating blade resects small portions of the fibroid, and these are suctioned into a tissue trap for pathologic evaluation. This technique alleviates the need for removal of fibroid “chips” from the cavity and it has been shown to be faster for trainees. 

Hysteroscopic tissue removal systems introduced an efficient, easy-to-use tool for hysteroscopic myomectomy. However, there are limitations, such as the high cost of the disposable element, as well as the difficulty resecting fundal fibroids and deep type 2 fibroids. In addition, dense and calcified fibroids can be very challenging to resect with these devices. One study showed that switching to a resectoscope in these cases allowed for finalization of the procedure [28]. However, another meta-analysis showed statistically significant improvement in complete resection of pathology when tissue extraction devices were used [29]. The surgeon should tactfully choose the best tool based on the pathology, its size, location, the patient’s goals, and the surgeon’s expertise.

The technique of surgery at the hospital is the same as that of the outpatient clinic, also with the physiological solute as distension media, with hospital surgery being more indicated for submucosal myoma. Due to the difficulty in fragmenting fibroids with smaller-caliber blades, surgery in the operating room, with instruments of greater caliber and power, makes myomectomy feasible with less operative time and improved efficiency [30,31].

Morcellators have expanded their use, with good acceptance, especially by those who are starting hysteroscopic surgery, due to practicality of use, short learning time, and non-use of energy (only mechanics), with good performance in the treatment of intracavitary lesions. Its limits are more intramural lesions, and lesions in cornual and fundic regions (Figure 11).

### 3.6. LASER Technique

The application of LASER will lead to myolysis with total destruction of the myoma or a late expulsion of it after volume reduction and ischemia. Therefore, it can be considered for nodules with a greater intramural component, in which surgery with a resectoscope could pose risks. The most common type of LASER used in hysteroscopy is the diode laser device, with a 5 Fr fiber, capable of mixing two different wavelengths, 980 nm and 1470 nm. A 980 nm wavelength is more absorbed by hemoglobin, leading to a higher coagulation effect. At 1470 nm, we will have a higher vaporization effect due to affinity to water. This mixing capacity allows combined effects that can be adjusted for each tissue and or surgery. In this deeper approach to the myometrium, the control of the free myometrial mantle should be monitored by Doppler ultrasound in order to avoid thermal damage to neighboring organs.

With the LASER fiber, the enucleation technique can be performed, incising the endometrium until reaching the pseudocapsule and then mobilizing the nodule mechanically (with another instrument), or waiting for its spontaneous expulsion or a second hysteroscopy with the resectoscope in 1 to 2 months, with intracavitary myoma (the OPPIuM Technique) [32] (Figure 12).

### 3.7. Radiofrequency Ablation Technique

This is also a myolysis technique, which is applied through ultrasound guidance. The system uses a handpiece for radiofrequency ablation, connected to an intrauterine ultrasound probe, forming a single integrated device so that the rods penetrate the myoma. This real-time ultrasound integration allows the physician to visualize and target as many fibroids as possible so they can be addressed [33].

### 3.8. Mazzon’s Cold Loop Technique

Mazzon’s technique was described in 1995 and is based on the resection of the submucosal component of the myoma using a resectoscope with a semicircle loop, with mono or bipolar energy, until reaching the intramural portion of the myoma. Upon reaching the pseudocapsule, the loop is changed to a more rigid one, which is not energized (cold loop), so that the myoma is mechanically mobilized until its enucleation. Then, a loop with energy returns to fragment and remove the myoma, which was left free in the uterine cavity. It has the advantage of approaching the myometrium without current, with lower risk of perforation and lower risk if perforation happens (thermal injury to other organs), with less thermal damage to the myometrium, less bleeding, and less intravasation [34] (Figure 13).

### 3.9. Mobilization and Enucleation Technique Using the Pseudocapsule—Lasmar

The technique published by Lasmar in 2002 has the name of “direct mobilization of the myoma”. It consists of incising the endometrium around the submucosal myoma using the resectoscope with the Collins loop until reaching the pseudocapsule, releasing the existing fibrous beams. Once the pseudocapsule is identified, with the same instrument, a movement similar to that performed in laparotomic and laparoscopic myomectomy is performed, separating the myoma from the myometrium in its entirety, causing it to slide into the myometrium.

As there is no traction, as in abdominal surgery, the base of the fibroid is released, starting at the lateral edges, entering with the Collins loop in the cervix–fundus direction without energy at the same time as slight mobilizations in the fibroid are made with the resectoscope assembly. The Collins loop is kept moving from the lateral to the central part of the myoma, parallel to the nodule and moving it with the hysteroscope, leading the myoma to progressively migrate to the uterine cavity until its complete release from the uterine wall. This is facilitated by the decompression of the myometrium, which, compressed by the growth of the nodule, progressively returns to its normal position by releasing the pseudocapsule, causing the intramural lesion to become intracavitary. This technique, like all those that perform myoma enucleation, has the same advantages: lower risk of perforation and risks associated with perforation (thermal injury to other organs), less thermal damage to the myometrium, and less bleeding and intravasation [35].

With the myoma totally in the cavity or almost totally, the nodule is sliced, using the Collins loop, in the longitudinal direction to remove it in large fragments, improving efficiency (Figure 14).

Sometimes, in the presence of large fibroids, it is difficult to mobilize the fibroid, and its release from the myometrium is not complete. In these cases, this great intracavitary portion of the myoma ends up touching the opposite wall, leaving no more space for progression, making it impossible to move. In these cases, fragmentation is necessary, even though the nodule is not completely free, but, even so, the safety level of the procedure is increased, since the largest portion of the nodule is already in the uterine cavity, with its migration from the myometrium deep to the surface.

With this technique, the limit of hysteroscopic myomectomy can be extended in relation to the measurement of the myometrial mantle before surgery from 10 to 3 mm [36].

The different techniques are important because not all hospitals have all the possibilities but knowing them, knowing how to use them, and knowing the best indication and limits are fundamental for those who are qualified in hysteroscopic surgery.

Regardless of the technique, some fibroids will not be removed in a single operative time; some procedures should be interrupted for safety, reinforcing the importance of preoperative assessment of the patient’s clinical conditions and classification of the fibroid, decisive data for knowledge, and risk prevention [20].

In incomplete myomectomy, a GnRH analogue can be prescribed for 2 to 3 months to cause the migration of the residual intramural component to the uterine cavity and, before the new surgical intervention, a second outpatient hysteroscopy and tests are performed to classify the myoma. In many cases, in the outpatient hysteroscopy itself, a myomectomy can be completed or the uterine cavity can be seen to be normal, as the myoma has been expelled [26].

Mainly in patients with infertility, outpatient second-look hysteroscopy is indicated 45 to 60 days after surgery to review the uterine cavity and lyse the adhesions, which may appear with the procedure and will be easily lysed with scissors or with the simple passage of the hysteroscope.

As operative bleeding is one of the most frequent risks in hysteroscopic myomectomy, the patient with severe anemia should not undergo surgery until the anemia has been corrected. Some treatments may be used preoperatively, mainly to block menstruation and hematologically recover the patient and others in the perioperative period to reduce intraoperative and postoperative bleeding.

## 4. Complications of Hysteroscopic Myomectomy

Among hysteroscopic surgeries, myomectomy is the one with the highest incidence of complications.

The incidence of complications in hysteroscopic myomectomy ranges from 0.8 to 2.6% [37,38].

Lasmar et al., in an international multicenter study, published an incidence of complications in 3.2% of 465 hysteroscopic myomectomies performed. Of the 15 patients with complications, 2 were fever, 2 had pain, 9 with bleeding, 1 uterine perforation, and 1 fluid overload syndrome [39].

**a—Laceration of the cervix** can occur at the time of dilation due to the positioning of the Pozi forceps and with the Hegar dilators, with dilation difficulty, especially in those who used GnRH before the procedure and in older patients. Revision of the laceration site, with tamponade and/or suturing of the area, has excellent results.

**b—Uterine perforation** can occur at the time of cervical dilatation or during surgery; when perforation occurs without the use of energy, only clinical observation, with the patient hospitalized for a few hours, is sufficient, as there will rarely be a need for surgical intervention. With the impossibility of uterine distention, the procedure must be suspended and the patient returns to the operating room in 3 months. However, if energy was used at the time of perforation, regardless of which, the indication for investigation of the pelvic and abdominal cavity is imperative, even with the great possibility of being negative. Laparoscopy or laparotomy may rule out bowel and/or bladder injuries. Bladder injury may be suspected in the presence of hematuria, as the patient with complex myomectomy has bladder catheterization for fluid balance. Hematuria will only happen when the anterior wall of the uterus is perforated, but it can happen lightly when the bladder catheter is moved, which should be evaluated with cystoscopy before considering laparoscopy.

Intestinal injury makes it more difficult to suspect without laparoscopy/laparotomy, especially thermal injuries that may take three days or more to fistulize, with potentially serious consequences, such as peritonitis and sepsis.

Vascular lesions can be suspected with hemodynamic instability.

Uterine perforation should be suspected in hysteroscopic myomectomy when there is very accelerated negative fluid balance (rapid escape of the distending medium) and vision of the uterine cavity cannot be established.

Attention is needed because “negative laparoscopy” may be justified, but the undiagnosed and untreated complication is not.

To reduce the possibility of uterine perforation at the time of cervical dilatation, some precautions should be taken:

1—Perform the bimanual exam to assess size, version, and uterine flexion.

2—Perform previous diagnostic hysteroscopy to identify the path and start dilatation with Hegar’s dilator # 4.

3—Remove the speculum after clamping the cervix with the Pozzi forceps and facilitate the rectification of the path.

4—Use dilators with a 0.5 cm diameter progression.

5—Limit with your index finger how much of the dilator will progress into the uterine cavity—the dilation is for the internal os only; there is no need to advance the Hegar dilator to the fundus of the uterus.

**c—Uterine bleeding** can happen due to the superficial vessels of the myomas or from the myometrial bed. The treatment, as previously described, consists of vessel coagulation, anti-hemorrhagic drugs, oxytocin, and placement of an intracavitary Foley catheter, with a well-distended balloon, for 4 to 12 h, always with patient monitoring.


**d—Fluid Overload**


Strict fluid balance is important, with great care from 1000 mL of negative balance, avoiding reaching 2000 mL. Rapid and massive absorption of fluids can lead to pulmonary edema, heart failure, encephalopathy, brain damage, seizures, coma, and death. When the distention medium is 1.5% glycine, massive absorption initially causes nausea, vomiting, and dizziness. Excess fluid in the intravascular space can lead to hemodilution, overload and heart failure, hyponatremia and increased ammonia, with encephalopathy, brain damage, and death. The severity of complications is directly associated with the volume absorbed in a short period of time. Prolonged surgical time can also increase absorption of the distention medium [40,41].

Some researchers use vasopressin and oxytocin to decrease the chance of intraoperative intravasation and bleeding, still awaiting further studies proving the effectiveness [42,43].

With the mannitol–sorbitol solution, massive absorption of fluid also causes hemodilution, which can lead to heart failure. As the condition is due only to hyperhydration, with no increase in plasma ammonia, encephalopathy is less frequent and less severe. It should be avoided in diabetic patients due to the possibility of hyperglycemia.

The use of saline and ringer lactate combined with bipolar current eliminates the possibility of electrolyte complications but not the risk of fluid overload and, consequently, heart failure.

**e—Infection** is not frequent in hysteroscopic surgeries; in myomectomy, it is possible due to the presence of residues, which could become infected, and the occlusion of the internal os, leading to the formation of hematometrium and pyometra. 

**f—Air embolism** is rare but it can be serious and fatal. Ambient air may be responsible, penetrating the venous circulation, during dilation of the cervical canal or through a solution of continuity in the myometrium, with greater risk with the patient in the Trendlenburg position, where the heart is below the level of the uterus. The risk of air embolism is similar in hysteroscopic myomectomy and other types of hysteroscopic surgery. The gas produced in bipolar vaporization with physiological solute is similar to that in monopolar vaporization with 1.5% glycine and does not appear to be responsible for causing embolism [44].


**g—Late complications**


Some late complications can occur, such as adhesions and placenta accreta, especially in areas of large resections. Some authors describe the incidence of adhesions after hysteroscopic myomectomy ranging from 1 to 13% [45]. Authors suggest the post-surgery hyaluronic acid intrauterine gel others the placement of a nonhormonal intrauterine device. What all services recommend is a review of the uterine cavity at 45 to 60 days postoperatively to review the uterine cavity and lysis of adhesions, especially in the patient who desires to conceive [46]. 

## 5. Clinical Outcomes of Hysteroscopic Myomectomy

Abnormal Uterine Bleeding (AUB):

Success rate of hysteroscopic myomectomy has been reported to be as high as 70–99%. Factors determining the success rate include the size, number, and location of the fibroids, in addition to the surgeon’s expertise and whether resection was complete or incomplete [47].

Fertility outcomes:

Although submucosal fibroids are frequently implicated in patients with subfertility and hysteroscopic myomectomy is commonly recommended, the literature is currently inconsistent in this regard. The Practice Committee of the American Society for Reproductive Medicine (ASRM) created guidelines that elaborate on the correlation between fertility and leiomyomas [16]. The heterogeneity of the study populations, type, location, number and size of fibroids, and the inclusion and exclusion criteria lead to difficulty drawing accurate conclusions that can guide practice recommendations [48,49,50].

## 6. Final Considerations

Hysteroscopic myomectomy is the most difficult and complex surgery among hysteroscopic surgeries, with potentially serious risks and complications. Its performance can be safe and efficient for the treatment of intrauterine disease, being the best therapeutic option for submucous myomas.

Safety in myomectomy is at two distinct moments, in the preoperative evaluation and in the operative act. In the preoperative period, it consists of the hemodynamic evaluation of the patient, the knowledge of the desire for a future pregnancy, and the classification of the uterine fibroid.

## Figures and Tables

**Figure 1 medicina-58-01627-f001:**
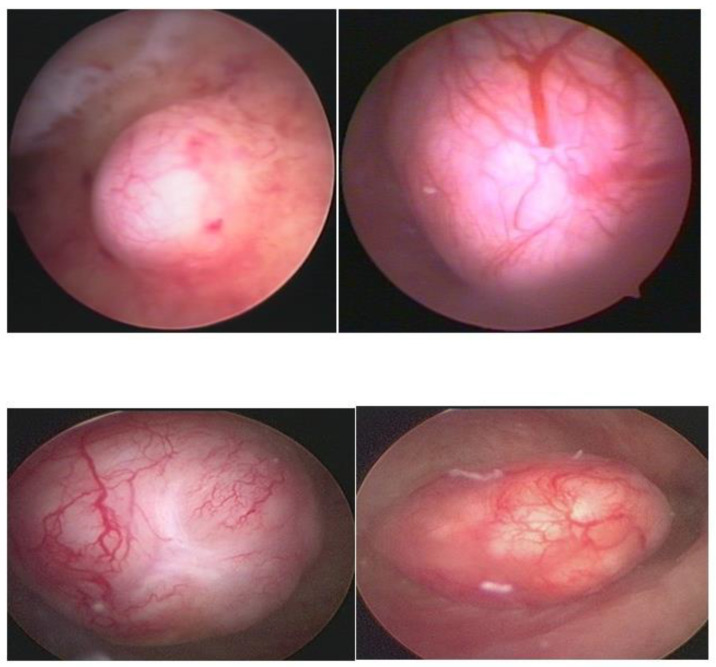
Submucous myoma—hysteroscopic view.

**Figure 2 medicina-58-01627-f002:**
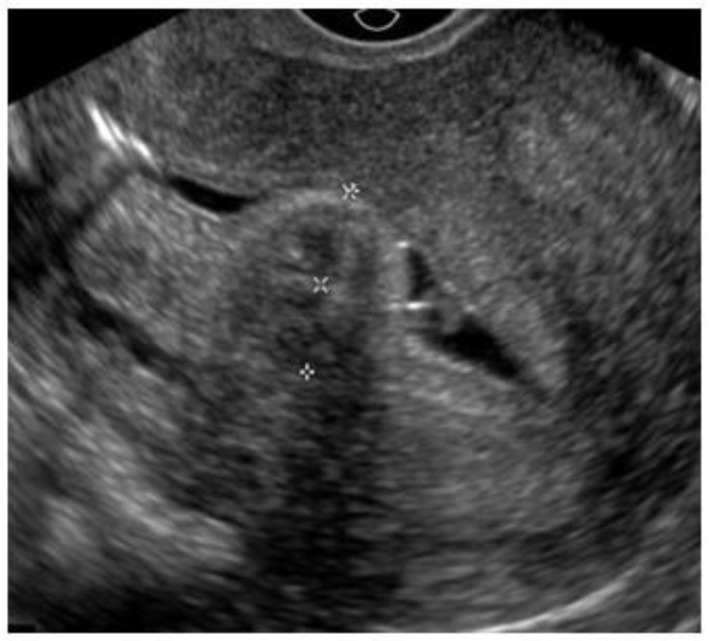
Submucous myoma on ultrasound. * Submucosal fibroid.

**Figure 3 medicina-58-01627-f003:**
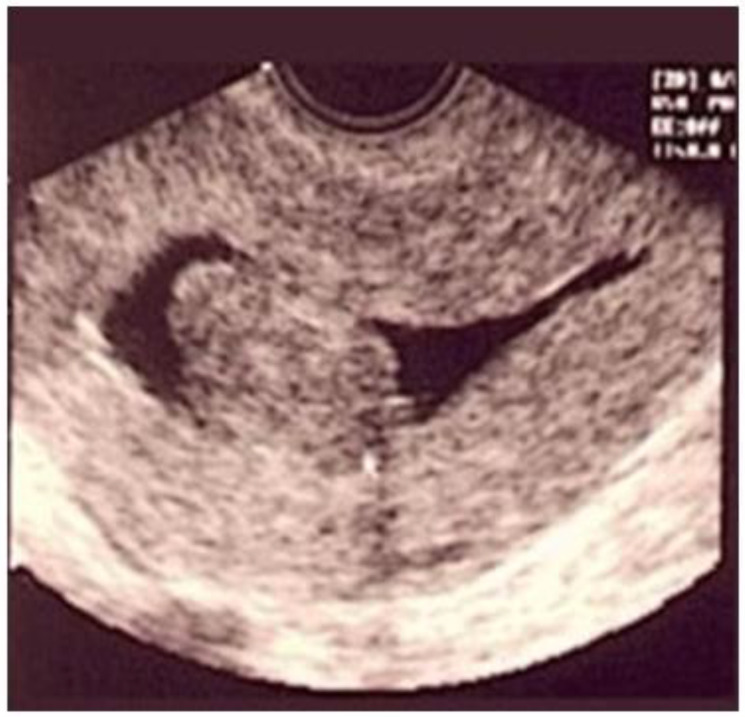
Submucous myoma in sonohysterography.

**Figure 4 medicina-58-01627-f004:**
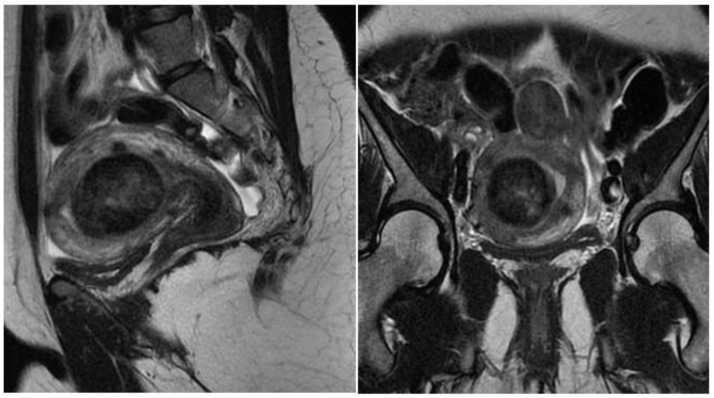
MRI with submucous myoma.

**Figure 5 medicina-58-01627-f005:**
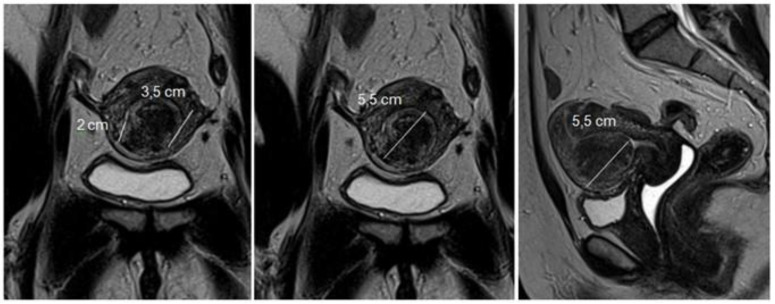
The effect of nearby fibroids.

**Figure 6 medicina-58-01627-f006:**
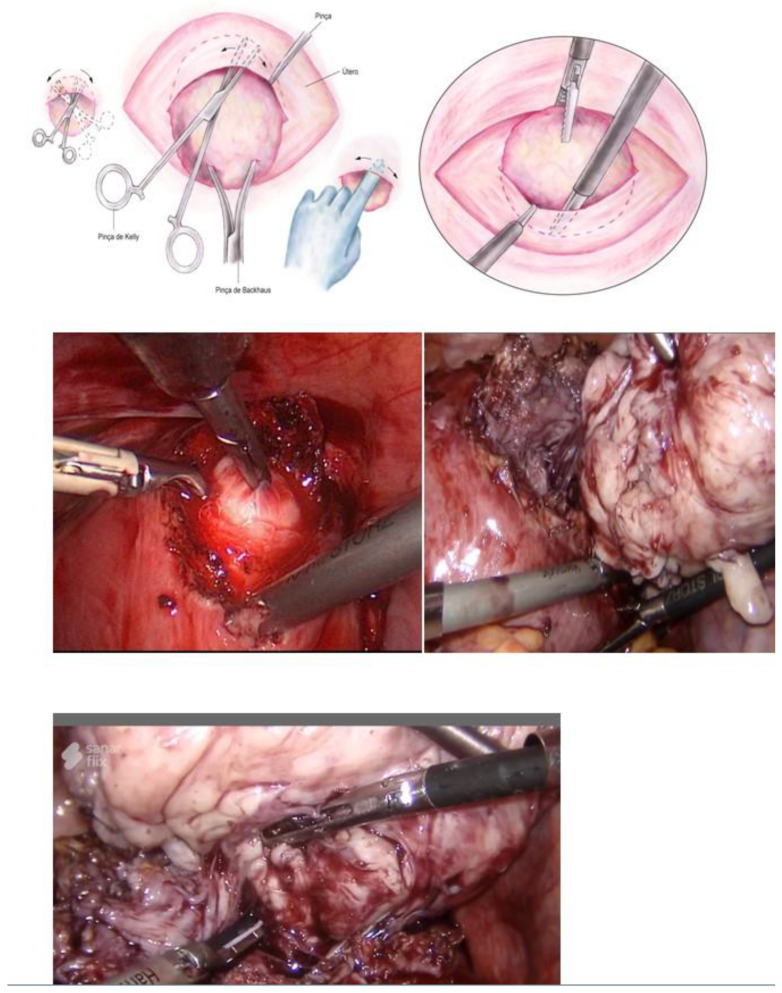
Laparotomic and laparoscopic myomectomy preserving pseudocapsule.

**Figure 7 medicina-58-01627-f007:**
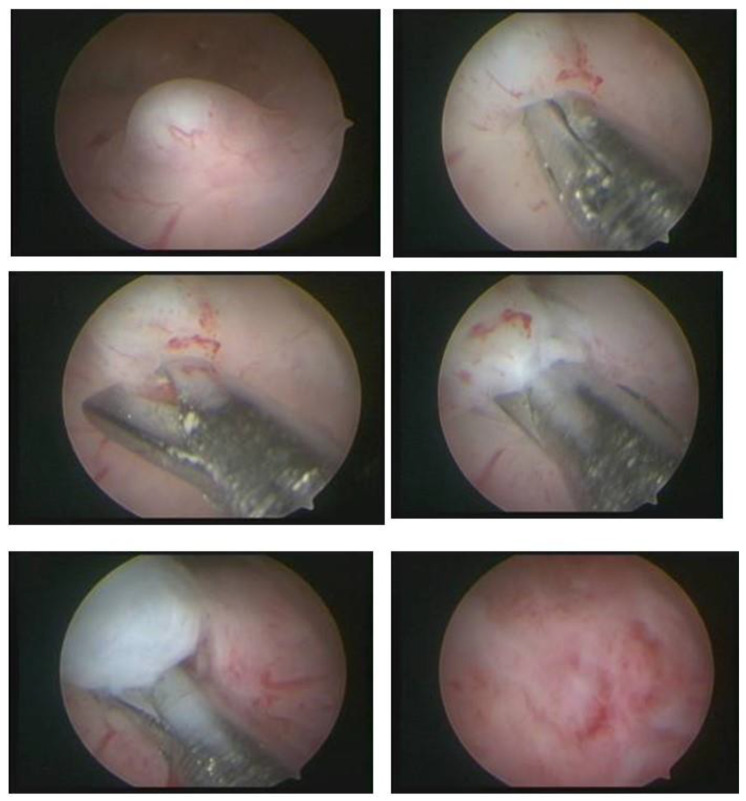
Office hystreroscopic myomectomy with scissor.

**Figure 8 medicina-58-01627-f008:**
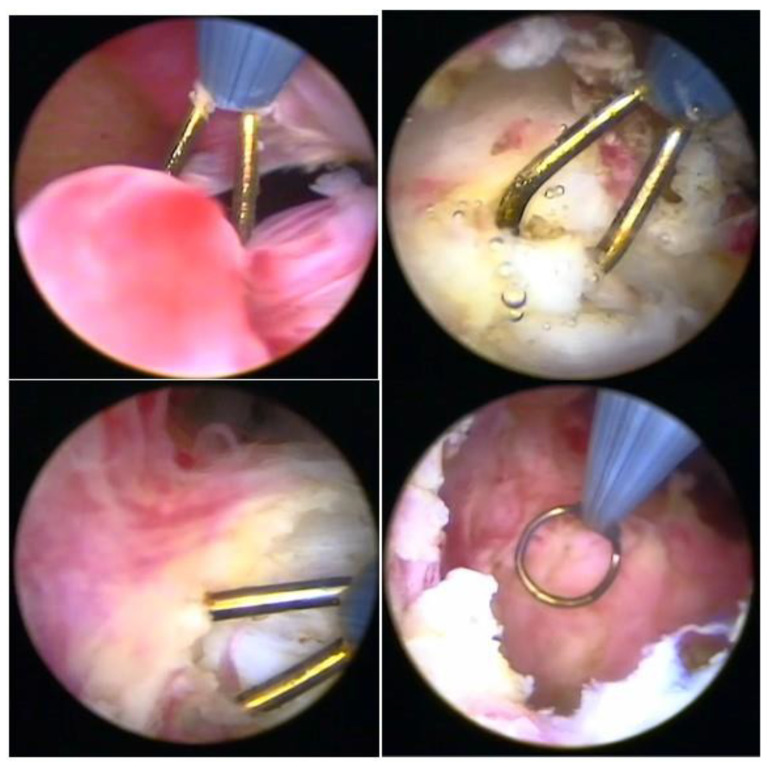
Office hysteroscopic myomectomy with miniresectoscope.

**Figure 9 medicina-58-01627-f009:**
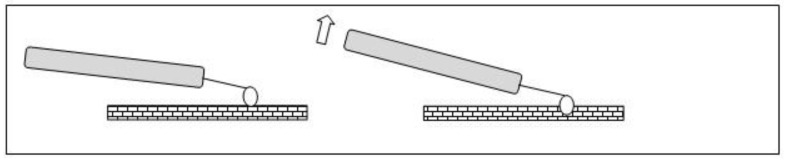
Correlation with hysteroscope angulation and tissue depth.

**Figure 10 medicina-58-01627-f010:**
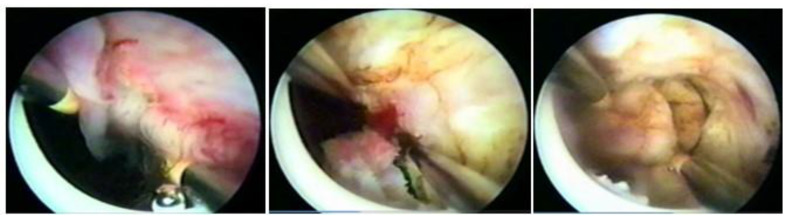
*Slicing* technique.

**Figure 11 medicina-58-01627-f011:**
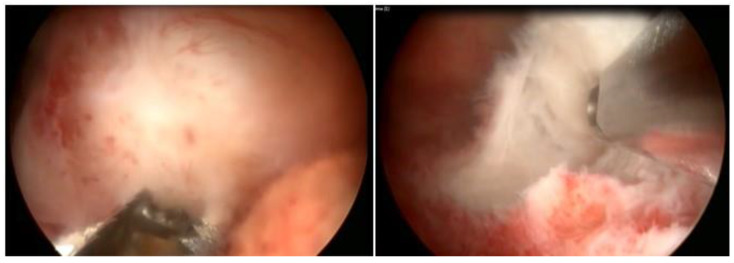
Morcellator technique—hysteroscopic mechanical tissue removal.

**Figure 12 medicina-58-01627-f012:**
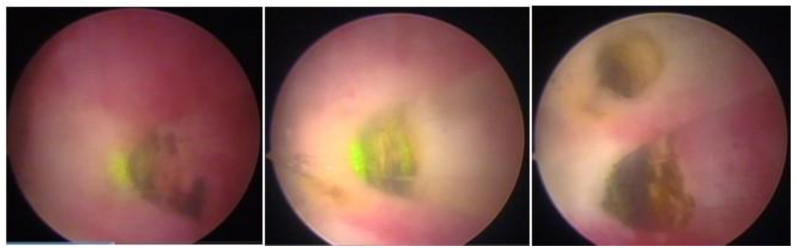
LASER in submucous myoma with deep intramural component.

**Figure 13 medicina-58-01627-f013:**
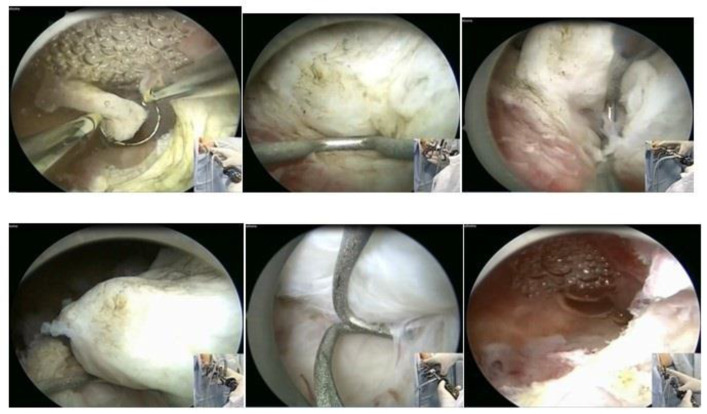
Mazzon technique with “cold loop”.

**Figure 14 medicina-58-01627-f014:**
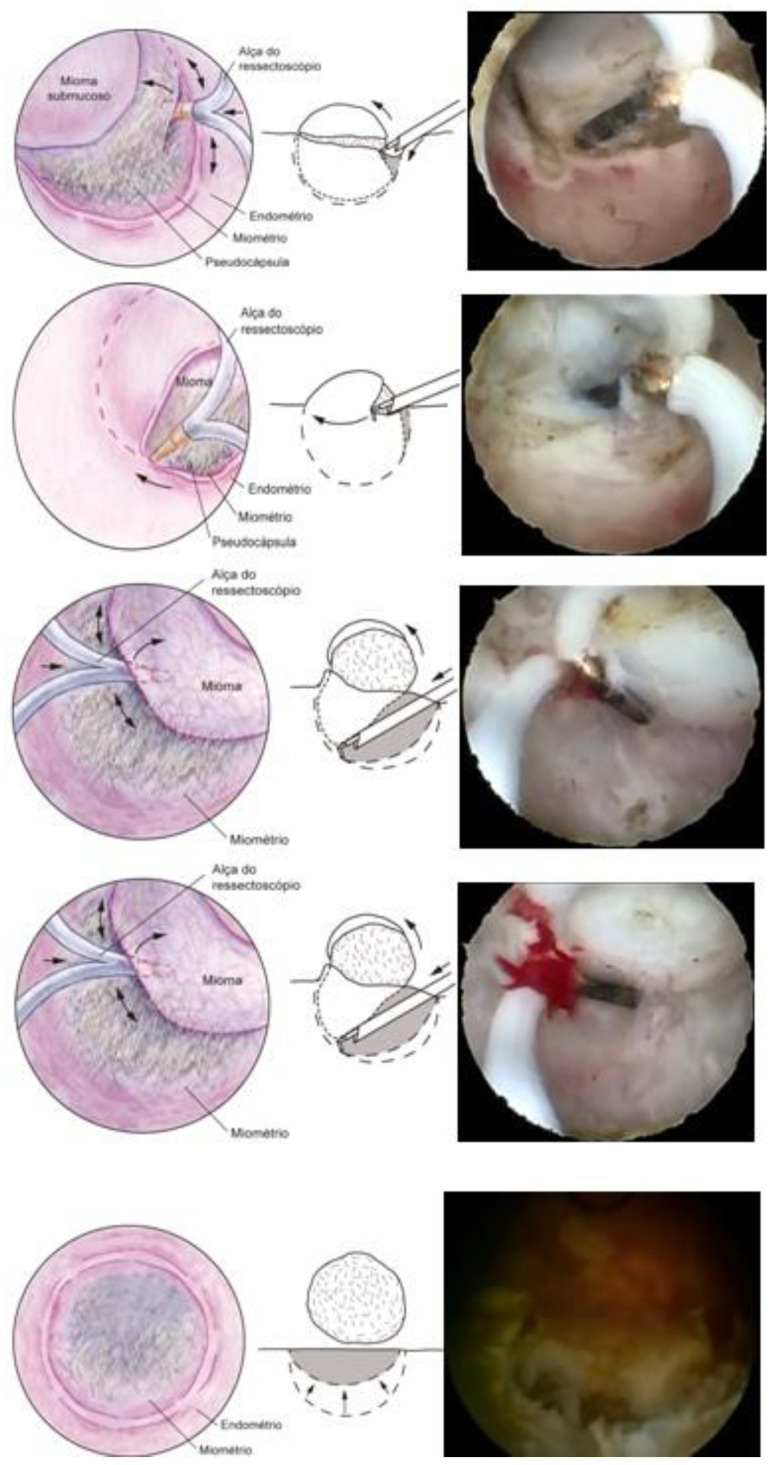
OR hysteroscopic myomectomy, with Collins electrode, enucleation, and preserving pseudocapsule.

**Table 1 medicina-58-01627-t001:** ESGE submucous myoma classification—Wamsteker (1993).

Type	Myoma Portions Inside of the Cavity
0	Totally inside
1	>50% inside
2	<50% inside

**Table 2 medicina-58-01627-t002:** Lasmar submucous myoma classification—STEPW classification (2005).

	Size	Penetration	Base	Topography	Lateral Wall	
**0**	**≤2 cm**	**0**	**≤1/3**	**lower**	**+1**	
**1**	**>2 to 5 cm**	**≤50%**	**>1/3 a 2/3**	**middle**		
**2**	**>5 cm**	**>50%**	**>2/3**	**upper**		
**Score**		**+**	**+**	**+**		
**Score**	**Group**	**Suggested Treatment:**				
**0 to 4**	**I**	**Low complexity hysteroscopic myomectomy.**				
**5 to 6**	**II**	**Complex hysteroscopic myomectomy, consider preparing with GnRH analog and/or 2-step surgery.**				
**7 to 9**	**III**	**Recommend an alternative non-hysteroscopic technique.**				
**Lasmar RB et al., J MinimInvasiveGynecol. July–August 2005; 12(4): 308–311** [18]						

## Data Availability

Not applicable.

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
