# Peer review of "HYSTEROSCOPIC MYOMECTOMY"

_medicina, 2022, doi:10.3390/medicina58111627_

Round 1

Reviewer 1 Report

The manuscript entitled “Laparoscopic myomectomy” represents a comprehensive approach to diagnosis, classification, and treatment of this benign, but very problematic gynecological pathology, especially in the reproductive period.

Above all, the authors made a good parallel between the two current classifications of submucosal myomas, ESGE and STEP W. Upon further analysis of the work, I notice that preference is given to the latter classification, which is more accessible and effective in practical work. Myomectomy techniques were also described in detail, with specific advantages and disadvantages, potential risks and complications, as well as the conditions for performing the aforementioned interventions.

Although the manuscript is of a review type, it has an exceptional educational character, especially important for specialists in gynecology and obstetrics and infertility subspecialists.

The manuscript posses great value and could be reconsidered for publication after minor revisions. The answers to the following questions may improve the quality of the manuscript. Please correct if possible.

My questions would be:

1. In the case of availability of all the mentioned techniques, what are the criteria for some of the mentioned techniques?

2. Whether years of life i.e. preserved/lost reproductive ability determine the type of operative approach?

3. Which technique is the best for preserving fertility, and removing fibroids that represent the potential causes of infertility?

4. What is the criterion for choosing the procedure in the case of multiple submucosal myomtous nodules below 30 mm individually?

5. How often are the mentioned techniques combined with non-hormonal conservative procedures for the treatment of fibroids (for example combination with Methotrexate)?

Author Response

The manuscript entitled “Laparoscopic myomectomy” represents a comprehensive approach to diagnosis, classification, and treatment of this benign, but very problematic gynecological pathology, especially in the reproductive period.

Above all, the authors made a good parallel between the two current classifications of submucosal myomas, ESGE and STEP W. Upon further analysis of the work, I notice that preference is given to the latter classification, which is more accessible and effective in practical work. Myomectomy techniques were also described in detail, with specific advantages and disadvantages, potential risks and complications, as well as the conditions for performing the aforementioned interventions.

Although the manuscript is of a review type, it has an exceptional educational character, especially important for specialists in gynecology and obstetrics and infertility subspecialists.
The manuscript posses great value and could be reconsidered for publication after minor revisions. The answers to the following questions may improve the quality of the manuscript. Please correct if possible.

My questions would be:

  1. In the case of availability of all the mentioned techniques, what are the criteria for some of the mentioned techniques?

Dear reviewer, thank you for your considerations and valuable comments.

Answer:  The main idea is to respect the pseudocapsule plane, regardless the technique that you prefer to use. If this plane is respected the patient will have less thermal damage, better cavity repair, less bleeding, less risk of overload, less risk of uterine perforation.

  1. Whether years of life i.e., preserved/lost reproductive ability determine the type of operative approach?

Answer: When the need for cavity integrity (infertility) is the main goal, the enucleation technique, respecting pseudocapsule is mandatory. In perimenopausal patients (no fertility issues), the concern about cavity is lower. As mentioned above, respecting the pseudocapsule has other advantages other than cavity preservation, such as decreasing myometrial damage and decreasing the risk of perforation, and should always be the default technique.

Which technique is the best for preserving fertility, and removing fibroids that represent the potential causes of infertility?

Answer: Respect the pseudocapsule, using energy sparingly, only at pseudocapsule fibers.

  1. What is the criterion for choosing the procedure in the case of multiple submucosal myomatous nodules below 30 mm individually?

 Answer: We should avoid treating myomas at opposite walls at the same time to decrease the risk of adhesions. Strict monitoring of the fluid balance is mandatory, and sometimes the procedure has to be terminated prematurely because of excessive fluid absorption.

  1. How often are the mentioned techniques combined with non-hormonal conservative procedures for the treatment of fibroids (for example combination with Methotrexate)?

Answer: The authors only have experience with hormonal treatment and GnRH agonists & antagonists. GnRH agonists have been shown to make the pseudocapsule plane more difficult to develop. The authors have no experience with using methotrexate for fibroids, and can’t reliably comment on its use.

Reviewer 2 Report

Lasmar et al review hysteroscopic myomectomy, a submission which could be given strong consideration for publication after the following items are addressed:

Major concerns

At line 43-44, fibroid growth or regression rate is described as “…9% over six months.” Such a claim must be rejected especially without qualifying the rate by patient age/race. The language should be toned-down and a reference provided.

At line 144, guidance is offered for appropriateness for MR imaging but with zero supporting literature—atypical for a review paper.

Line 225: What is ‘high success rate’? - meaningless without definition.

Line 269-71 discusses procedure safety but again no details (or references) are shared.

Laser ablation/myolysis (beginning at line 426) needs information on instrument details. Multiple types of lasers are used in GYN surgery, each with advantages & drawbacks. This section assumes the reader somehow knows which device is being discussed, a confusing feature.

How is this review different from their closely related publication which appeared in Minim Invasive Ther Allied Technol 2021;30(5):263?

Minor issues

Figures 5 & 6 are actually tables. They appear to be inserted PDFs/jpegs of a draft word document. Using that format puts editorial red lines throughout (should be removed).

Line 230: Who speaks for everyone? Phrasing should be revised.

Line 238-242: This entire paragraph should be closely edited and condensed. Similarly, the odd sentence at lines 243-5 could be dropped entirely with no real loss of information.

Poor organization - Morcellation is first introduced at line 240, but then no follow-up until line 392. Why?

Check reference formatting for consistency and compliance w/journal style.

Author institutional affiliations are not job postings or CV entries. This should be limited to clinics, hospitals, universities, etc as locations where research/review was conducted.

Author Response

Reviewer# 2:

Lasmar et al review hysteroscopic myomectomy, a submission which could be given strong consideration for publication after the following items are addressed:

Major concerns

At line 43-44, fibroid growth or regression rate is described as “…9% over six months.” Such a claim must be rejected especially without qualifying the rate by patient age/race. The language should be toned-down and a reference provided.

Answer: Thanks for pointing this out. This was corrected in the text

At line 144, guidance is offered for appropriateness for MR imaging but with zero supporting literature—atypical for a review paper.

Answer: inserted supporting reference

Line 225: What is ‘high success rate’? - meaningless without definition.

Answer: the text was modified

Line 269-71 discusses procedure safety but again no details (or references) are shared.

Answer: inserted a reference

Laser ablation/myolysis (beginning at line 426) needs information on instrument details. Multiple types of lasers are used in GYN surgery, each with advantages & drawbacks. This section assumes the reader somehow knows which device is being discussed, a confusing feature.

Answer: Thank you for highlighting this. This was modified in the text

How is this review different from their closely related publication which appeared in Minim Invasive Ther Allied Technol 2021;30(5):263?

Answer: here we focused Only on myomectomy techniques, bringing new figures and discussing the techniques more extensively.

Minor issues

Figures 5 & 6 are actually tables. They appear to be inserted PDFs/jpegs of a draft word document. Using that format puts editorial red lines throughout (should be removed).

Answer: Thank you.  This was modified in the text

Line 230: Who speaks for everyone? Phrasing should be revised.

Answer: modified in text

Line 238-242: This entire paragraph should be closely edited and condensed. Similarly, the odd sentence at lines 243-5 could be dropped entirely with no real loss of information.

Answer: modified in text

Poor organization - Morcellation is first introduced at line 240, but then no follow-up until line 392. Why?

Answer: We started with the standard enucleation technique, in office and in the OR. Fragmentation techniques were adressed later on.

Check reference formatting for consistency and compliance w/journal style.

Answer: Thank you for pointing this out. This was modified in the text

Author institutional affiliations are not job postings or CV entries. This should be limited to clinics, hospitals, universities, etc as locations where research/review was conducted.

Answer: modified in text

Dear reviewer, thanks for your considerations.

Reviewer 3 Report

This is interesting Review manuscripts on hysteroscopic myomectomy. The authors described in detail the procedure and preoperative assessment of fibroids. They emphasize precise myomas diagnosis as the most important before the procedure.

However, the authors should consider the following recommendations:

There are no references in section Preoperative assessment lines 104-151. I suggest to add the references.

Also in sections:  Hospital hysteroscopic myomectomy (lines 319-362) and Radiofrequency ablation technique (lines 440-445) there is no literature data.

Lines 501-510- please add the references.

In section Pre-operative classification it is worth to cite a Consensus Statement from the Global Congress on Hysteroscopy Scientific Committee (DOI: 10.1016/j.jmig.2018.06.020)

Figures are interesting and complement the text.

References should be supplemented with new items, according to above suggestions, and should be written in accordance with the Instructions for authors:

Journal Articles:
1. Author 1, A.B.; Author 2, C.D. Title of the article. 
Abbreviated Journal Name YearVolume, page range.

I suggest also citing more recent articles about the topic.

Author Response

Reviewer# 3:

This is interesting Review manuscripts on hysteroscopic myomectomy. The authors described in detail the procedure and preoperative assessment of fibroids. They emphasize precise myomas diagnosis as the most important before the procedure.

However, the authors should consider the following recommendations:

  1. There are no references in section Preoperative assessment lines 104-151. I suggest to add the references.

Answer: The authors wish to thank the reviewer for bringing this up to their attention. This was modified in the text.

  1. Also in sections: Hospital hysteroscopic myomectomy (lines 319-362) and Radiofrequency ablation technique (lines 440-445) there is no literature data.

Answer: modified in text

  1. Lines 501-510- please add the references.

Answer: modified in text

  1. In section Pre-operative classification it is worth to cite a Consensus Statement from the Global Congress on Hysteroscopy Scientific Committee (DOI: 10.1016/j.jmig.2018.06.020)

Answer: modified in text

  1. Figures are interesting and complement the text.

Answer: Thank you

  1. References should be supplemented with new items, according to above suggestions, and should be written in accordance with the Instructions for authors:

Journal Articles:

Author 1, A.B.; Author 2, C.D. Title of the article. Abbreviated Journal Name Year, Volume, page range.

Answer: modified in text

  1. I suggest also citing more recent articles about the topic.

Answer: modified in text

Dear reviewer, thanks for your considerations.
